# Comparative Characterization of Different Molecular Formats of Bispecific Antibodies Targeting EGFR and PD-L1

**DOI:** 10.3390/pharmaceutics14071381

**Published:** 2022-06-29

**Authors:** Nishant Mohan, Atul Agrawal, Yi Shen, Katie L. Winarski, Yukinori Endo, Milos Dokmanovic, Deborah Schmiel, Jiwen Zheng, David S. Rotstein, Lorraine C. Pelosof, Wen Jin Wu

**Affiliations:** 1Division of Biotechnology Review and Research 1, Office of Biotechnology Products, Office of Pharmaceutical Quality, Center for Drug Evaluation and Research, U.S. Food and Drug Administration, Silver Spring, MD 20993, USA; nishant.mohan@fda.hhs.gov (N.M.); atul30sls@gmail.com (A.A.); yi.shen@fda.hhs.gov (Y.S.); katie.winarski@gmail.com (K.L.W.); yukinori.endo@fda.hhs.gov (Y.E.); milos.dokmanovic@fda.hhs.gov (M.D.); deborah.schmiel@fda.hhs.gov (D.S.); 2Division of Biology, Chemistry and Materials Science, Office of Science and Engineering Laboratories, Center for Devices and Radiological Health, U.S. Food and Drug Administration, Silver Spring, MD 20993, USA; jiwen.zheng@fda.hhs.gov; 3Division of Compliance, Office of Surveillance and Compliance, Center for Veterinary Medicine, U.S. Food and Drug Administration, Derwood, MD 20855, USA; david.rotstein@fda.hhs.gov; 4Division of Oncology 3, Office of Oncologic Diseases, Center for Drug Evaluation and Research, U.S. Food and Drug Administration, Silver Spring, MD 20993, USA; lorraine.pelosof@fda.hhs.gov

**Keywords:** bispecific antibodies (BsAbs), EFGR, PD-L1, triple-negative breast cancer (TNBC), bioassays

## Abstract

We generated two IgG1-like bispecific antibodies (BsAbs) with different molecular formats, symmetrical DVD-Ig and asymmetrical knob-in-hole (KIH), targeting the same antigens, EGFR and PD-L1 (designated as anti-EGFR/PD-L1). We performed the physiochemical and biological characterization of these two formats of anti-EGFR/PD-L1 BsAbs and compared some key quality attributes and biological activities of these two formats of BsAbs. Physiochemical binding characterization data demonstrated that both formats bound EGFR and PD-L1. However, the binding affinity of the KIH format was weaker than the DVD-Ig format in Biacore binding assays. In contrast, both DVD-Ig and KIH BsAbs had similar ELISA and cell surface binding activities, comparable to mAbs. Triple-negative breast cancer (TNBC) cells and a xenograft model were used to test the potency of BsAbs and other biological activities. Results showed that anti-EGFR/PD-L1 BsAbs exhibited in vitro and in vivo antitumor proliferation activity, but there was a difference in the potencies of the respective BsAb formats (DVD-Ig and KIH) when different cells or assays were used. This study provides evidence that the potency of the BsAbs targeting the same antigens can be affected by the respective molecular features, and selection of appropriate cell lines and assays is critically important for the assay development and potency testing of BsAbs.

## 1. Introduction

Programmed death-ligand 1 (PD-L1), an immune checkpoint protein expressed on cancer cells and immune cells, interacts with its binding partner programmed cell death-1 (PD-1) to inhibit T cell proliferation and cytokine production [1]. Upregulation of PD-L1 is an adaptive immune mechanism exploited by cancer cells to evade the antitumor immune response [2,3]. It is now well established that PD-L1 is overexpressed in triple-negative breast cancers (TNBC), a subtype of breast cancer characterized by the lack of expression of the estrogen receptor (ER), progesterone receptor (PR) and human epidermal growth factor 2 receptor (HER2) [4,5]. Atezolizumab, a monoclonal antibody (mAb) targeting PD-L1, demonstrated promising antitumor activity in pre-clinical models of TNBC [4,6]. Based on the outcome of a phase 3 IMpassion130 trial which demonstrated that an atezolizumab plus nab-paclitaxel clinical regimen prolonged progression-free survival among patients with metastatic triple-negative breast cancer, atezolizumab received accelerated FDA approval in March 2019 for patients with triple-negative breast cancer (TNBC) whose tumors express PD-L1 [7,8]. Results from recent phase 3 IMpassion131 study did not show a significant benefit from atezolizumab and paclitaxel combinations in advanced TNBC tumors which lead to withdrawal of atezolizumab for the treatment of TNBC patients [9]. Withdrawal of atezolizumab from clinical regimens for TNBC patients warrants further research into improving currently available treatment options and investigating novel strategies to potentiate antitumor immune activity of atezolizumab in TNBC models.

Aberrant activation of epidermal growth factor receptor (EGFR) signaling pathways promoting tumor growth and progression has been extensively reported in a majority of human cancers [10]. Although EGFR is overexpressed in about 50% of TNBC cases, anti-EGFR therapies have produced disappointing clinical outcomes for TNBC patients, therefore, none of the anti-EGFR targeting mAbs (e.g., cetuximab and panitumumab) or tyrosine kinase inhibitors (e.g., erlotinib and gefitinib) received regulatory approval for TNBC indication [11]. There is a strong rationale for the co-targeting of EGFR and PD-L1 in advanced and metastatic TNBC. Cetuximab exhibits its antitumor activity by inhibiting EGFR-mediated tumor growth and stimulating NK-cell-mediated antibody-dependent cellular cytotoxicity (ADCC) activity to lyse tumor cells [12]. Despite showing strong antitumor activity in EGFR-expressing TNBC pre-clinical models, cetuximab failed to exhibit significant clinical benefit in TNBC patient populations [13,14]. On the other hand, atezolizumab blocks the interaction of PD-L1 expressed on tumor cells and tumor-infiltrating immune cells with both PD-1 and B7.1 which results in reduced immunosuppressive signals and enhanced T cell-mediated antitumor immunity [15,16]. PD-L1 expression can be regulated by EGFR signaling and significant crosstalk between EGFR signaling and PD-L1 signaling has been reported [17,18]. Therefore, combining the antitumor mechanisms of these two mAbs using a uniquely designed bispecific antibody (BsAb) could provide a more potent killing of TNBC cells and could be developed as an attractive therapeutic drug to treat patients with TNBC.

BsAbs are an emerging class of therapeutic molecules that have capacity to simultaneously target two antigens and provide a unique opportunity to combine two target functionalities using single antibody-based molecules [19,20]. The diversity of BsAb structure is fast-growing, which creates a large amount of BsAb formats and provides great functional variety. However, it also poses significant challenges to physiochemical and biological characterization of BsAbs to support clinical development. We designed and generated two different IgG-like molecular formats of BsAbs, a symmetric tetravalent dual-variable domain antibody (DVD-Ig) format and an asymmetric bivalent knob-in-hole (KIH) format based on two widely used BsAb manufacturing platforms. Both BsAbs target the same antigens, PD-L1 and EGFR. In this investigation, we performed comparative characterization of physiochemical and biological properties of these two formats of anti-EGFR/PD-L1 BsAbs and investigated their antitumor potency and activity using TNBC cellular and xenograft models.

## 2. Materials and Methods

### 2.1. Cell Lines and Culture

The TNBC cell lines MDA-MB-231, BT-20 and MDA-MB-468 were obtained from American Type Culture Collection (ATCC, Manassas, VA, USA) and propagated at 37 °C under 5% CO_2_ as described previously [21,22]. Expi293F cells (cat# A14527 ThermoFisher Scientific, Waltham, MA, USA) are human kidney cells derived from 293F cell line and were maintained in the suspension culture using Expi293 Expression Medium. Therapeutic monoclonal antibodies cetuximab (Erbitux^®^) and atezolizumab (Tecentriq^®^) were purchased from the FDA-designated pharmacy. The EGF, recombinant PD-L1 and recombinant EGFR were obtained from RayBiotech (Peachtree Corners, GA, USA).

### 2.2. Construction, Expression and Purification of the Anti-EGFR/PD-L1 BsAb

The amino acid sequences of anti-EGFR mAb (cetuximab) and anti-PD-L1 mAb (atezolizumab) were obtained from publicly available International Immunogenetics Information System (http://www.imgt.org) database accessed on 27 May 2017. The pcDNA3.1 (+) vector plasmids expressing heavy chains and light chains of DVD-Ig and KIH formats of anti-EGFR/PD-L1 BsAbs were generated by Genscript (Piscataway, NJ, USA) using standard DNA recombinant technologies. The two plasmids of DVD-Ig format and three plasmids of KIH format anti-EGFR/PD-L1 BsAbs encoding heavy and light chains were transiently co-transfected into Expi293 cells using an Expifectamine 293 Transfection Kit (Thermo Fisher Scientific, Rockford, IL, USA) as per manufacturer’s instructions. The purification of both formats of anti-EGFR/PD-L1 BsAbs was performed by affinity chromatography using immobilized protein-A agarose resin as described previously [23]. The production yield of anti-EGFR/PD-L1 DVD-Ig and KIH formats is approximately 0.05–0.2 mg per ml from 125 mL of culture supernatant. The BsAb protein samples were formulated in PBS as storage buffer and stored at −80 °C until further use.

### 2.3. Biacore Binding Assay

Biacore T200 optical biosensor (GE Healthcare, Piscataway, NJ, USA) was used to assess binding affinity of anti-EGFR/PD-L1 BsAbs using a CM5 sensor chip. Briefly, carboxyl groups on the CM5 sensor chip surface were activated using an amine coupling kit. Protein A at a concentration of 20 μg/mL in 10 mM sodium-acetate buffer, pH 4.5, was allowed to flow over the chip surface to obtain response units of reacted protein (>200 RU). Binding kinetics were measured by passing the mAbs and BsAbs at 8 µg/mL and various concentrations of antigens (human EGFR and PD-L1 protein) over the chip surface for 3 min. The dissociation of bound analytes was monitored while the surface was washed for 10 min. Remaining analytes were removed with two 30 s injections of 10 mM glycine-HCl, pH 1.5. The kinetic parameters were determined by collectively fitting the overlaid sensograms locally using the BIAevaluation 4.1 software (Uppsala, Sweden) to the 1:1 Langmuir binding model.

### 2.4. ELISA and Cell Surface Binding Assays

The standard ELISA binding assay, cell surface binding assays using flow cytometry and coimmunoprecipitation assay were performed to detect the binding of mAbs and anti-EGFR/PD-L1 BsAbs to receptors EGFR or PD-L1 as described previously [23].

### 2.5. CellTiter-Glo Luminescent Cell Viability Assay

The assay was performed according to manufacturer instruction (Promega cat# 7570).

### 2.6. PD-1/PD-L1 Blockade Assay

The PD-1/PD-L1 Blockade Bioassay (Promega, cat# J1250) is a bioluminescent cell-based assay, in which two genetically engineered cell lines, PD-1 effector cells and PD-L1 aAPC/CHO-K1 cells, are cultured together to block the PD-L1 and PD-1 interaction which inhibits T cell receptor (TCR) activation. When PD-1 or PD-L1 blocking mAbs are added into the co-culture system, PD-L1/PD-1 interaction is impaired and the TCR is activated which can be detected by adding Bio-Glo™ Reagent. The assay was performed in a 96-well plate format according to manufacturer’s instructions. Briefly, PD-L1 aAPC/CHO-K1 cells were cultured in Ham’s F12 media containing 10% FBS in a white, flat bottom 96-well plate and incubated overnight in a 37 °C incubator with 5% CO_2_. The next day, the supernatant was discarded and two-fold serial dilutions of the parental mAbs and BsAbs (ranging from 0.02 µg/mL to 10 µg/mL) were added into the plates containing the PD-L1 aAPC/CHO-K1 cells. Then, a vial of PD-1 effector cells was thawed in RPMI media containing 1% FBS and added into the plates. After 4–5 h incubation, Bio-Glo™ Reagent was added into the plates and luminescence was measured using the Promega Glomax Discover plate reader (cat# GM3000). The assay was performed in triplicate and relative luminescence units (RLU) were plotted against antibody concentrations.

### 2.7. ADCC Reporter Bioassay

The ADCC Reporter Bioassay developed by Promega (cat# G7010) is a modified version of the classic ADCC assay in which Jurkat effector cells are engineered to stably express the FcγRIIIa receptor, V158 (high affinity) variant, and an NFAT (nuclear factor of activated T cells) response element driving expression of firefly luciferase. Luciferase produced as a result of NFAT pathway activation by engineered Jurkat effector cells is quantified with luminescence readout. The ADCC reporter bioassay was performed in a 96-well plate format according to manufacturer’s instructions. Briefly, target cells MDA-MB-468 were seeded in a white-bottom 96-well plate at 10,000 cells/well in 100 µL volume using the ADCC Bioassay buffer as described in the manufacturer’s kit. The next day, serially diluted antibodies (ranging from 0.02 µg/mL to 10 µg/mL) were added into wells. ADCC effector cells were thawed and prepared as indicated in the manufacturer’s protocol and then added to the target cells at an effector to target ratio of 6:1. After 5–6 h of incubation in the cell culture incubator, Bio-Glo™ Luciferase Assay Reagent was added to the plates and luminescence was measured using the GloMax discover plate reader.

### 2.8. T Cell-Mediated Killing Assays

T cell-mediated killing of tumor cells was performed as described previously [4]. Briefly, MDA-MB-231 and BT-20 cancer cells were seeded in a 24-well plate in media containing 10% FBS and allowed to adhere overnight. The next day, cancer cells were washed with PBS and then labeled with calcein AM cell-permeant dye (ThermoFisher Scientific, cat# C1430), which generates green fluorescence in live cells after acetoxymethyl ester hydrolysis by intracellular esterases. After labelling with calcein AM dye for 30–40 min, cancer cells were washed with PBS and then co-incubated with T cells alone or T cells with mAbs and BsAbs for 24 h. The antibody concentrations used in this experiment were 20 µg/mL for Cetux, ATE and DVD-Ig, and 40 µg/mL for KIH format of BsAbs. After overnight incubation, culture plates were washed with PBS to remove residual T cells, and calcein AM-labeled cancer cells were counted using the Celigo cell imaging system (Nexcelom Bioscience, Lawrence, MA, USA). T cells were isolated from PBMCs of healthy donors obtained from fresh leukopaks [National Institutes of Health (NIH) Blood bank] using the EasySep Human T cell isolation kit (STEMCELL Technologies, Vancouver, BC, Canada, cat# 17951). Briefly, PBMC (50 × 10^6^) were treated with ACK Lysing buffer (Lonza, Basel, Switzerland, cat# BP10-548E) and then washed twice with PBS. After centrifugation, cells were resuspended in Robosop buffer (STEMCELL Technologies cat# 20104) in a sterile 5 mL tube and then 75 µL of antibody cocktail was added with gentle mixing. The tube was placed in a magnetic stand for 5 min to allow negative selection and then T cells were carefully transferred to a 15 mL tube. T cell activator cocktail (STEMCELL Technologies cat# 10970), which binds to and crosslinks CD3, CD28 and CD2 cell surface ligands, was added into the T cell culture media to provide primary and co-stimulatory signals for T cell activation.

### 2.9. Detection of Cytokine Release

The production of IFN and TNF was measured in a co-culture of human T cells mixed with MDA-MB-231 and BT-20 cells in presence of parental mAbs and BsAbs. Briefly, MDA-MB-231 and BT-20 were seeded in 24-well plates and allowed to adhere overnight. The next day, activated human T cells isolated from PBMC were added at a 1:3 ratio along with indicated treatments (20 µg/mL for Cetux, ATE and DVD-Ig BsAbs, and 40 µg/mL for KIH format of BsAbs). After 24 h of incubation, cell culture supernatant was collected and then subjected to the human IFNγ Quantikine ELISA Kit (R & D Systems, Minneapolis MN, USA, cat# DIF50C) for IFN measurement and Human TNFα Quantikine ELISA Kit (R & D Systems, Minneapolis, MN, USA, cat# DTA00D) for TNF measurement according to instructions provided by the manufacturer.

### 2.10. SDS-PAGE and Western Blotting

For non-reduced SDS-PAGE analysis of mAbs and BsAbs, 2 µg of protein samples were prepared in 2× Laemmli sample buffer (Biorad, Hercules, CA, USA, cat# 1610737). 2-mercaptoethanol was added into sample buffer as a reducing agent for reduced SDS-PAGE. The protein samples were then boiled for 5 min. After boiling, protein samples were separated using 4–15%-gradient SDS-PAGE gel, stained with SimplyBlue coomassie stain (ThermoFisher Scientific, cat# LC6065) for 1 h, washed with deionized water and then visualized with ChemiDoc MP gel imaging system (BioRad). For western blotting, cells were lysed in NP-40 lysis buffer and whole-cell lysates (WCL) were prepared in sample buffer (BioRad) containing reducing agent and subjected to 4–15%-gradient SDS-PAGE gel separation as described previously [24]. After separation, gel was transferred to PVDF membrane, blocked with 5% non-fat milk in TBST and then incubated with primary antibodies overnight at 4 °C. Protein bands were visualized by ChemiDoc MP gel imaging system (BioRad). The primary antibodies directed against EGFR, phospho-EGFR, ERK1/2 (p44/42 MAPK), phospho-ERK1/2, Akt and phospho-Akt were purchased from Cell Signaling Technologies. The actin antibody and HRP-conjugated secondary antibodies were purchased from Sigma-Aldrich (St. Louis, MO, USA).

### 2.11. Tumor Xenograft Study

The athymic nude tumor xenograft study using MDA-MB-231 cells was performed as described previously [23,25]. All animal experiments were performed in accordance with animal protocol #WO-2018-26, which was approved by the United States Food and Drug Administration (FDA) Institutional Animal Care and Use Committee, in accordance with the U.S. Public Health Service Policy on Humane Care and Use of Laboratory Animals. Female athymic nude mice at the age of 4–6 weeks were obtained from Jacksons Lab (Bar Harbor, ME, USA) and subcutaneously injected with 5 million MDA-MB-231 cells on the left and right flanks. After tumor volume reached approximately 50–100 mm^3^, all tumor-bearing mice were randomly assigned into 5 treatment groups comprising of saline, Cetux, ATE, DVD-Ig and KIH. The mice were treated with mAbs and BsAbs intraperitoneally at 10 mg/kg dose twice a week for approximately 4 weeks. Tumor dimensions (length and width) were measured using digital calipers twice a week in all treatment groups and tumor volume was calculated using following formula: length × (width)^2^ × 0.5 where length was the longest distance and width was shortest distance. To monitor systemic toxicity caused by saline and treatment groups, the body weight of mice was measured twice a week. After completion of treatment course on day 29, the mice in all the treatment groups were euthanized and tumors were excised from mice.

### 2.12. Statistical Significance

The statistical analysis was performed using Microsoft Excel and GraphPad Prism software and data were compared by a two-tailed Student’s *t*-test. The data was presented in the form of mean ± standard deviation and error bars represent standard deviation. The differences between two groups were considered statistically significant when *p* < 0.05 (*, *p* < 0.05; **, *p* < 0.01).

## 3. Results

### 3.1. Generation of DVD-Ig and KIH Formats of Anti-EGFR/PD-L1 BsAbs

Two different strategies were employed to generate anti-EGFR/PD-L1 BsAbs. For the symmetrical DVD-Ig format of anti-EGFR/PD-L1 BsAb, single-chain heavy variable fragments of cetuximab (Cetux) and atezolizumab (ATE) were linked to human IgG1 backbone to generate heavy chain. The light chain of the DVD-Ig format consists of light-chain variable regions of ATE and Cetux connected to human Ig kappa constant regions (Figure 1A). The asymmetrical KIH format of anti-EGFR/PD-L1 BsAbs consists of two half-heavy chains containing “knob” and “hole” mutations. In KIH technology, a “knob” is created by replacing a small amino acid in CH3 domain with a larger one, and a “hole” is created by mutating the larger amino acid with a smaller one, while allows heavy-chain heterodimerization. In “knob” heavy chain of anti-EGFR/PD-L1 BsAbs, the variable region of Cetux heavy chain is connected to the human IgG1 constant region containing T350V_L351Y_F405A_Y407V mutations. In the “hole” heavy chain of anti-EGFR/PD-L1 BsAbs, the scFv of atezolizumab is connected to human IgG1 Fc containing T366L_K392L_T394W mutations. Knob- and hole-variant mutations used for KIH BsAbs were optimized for stability of KIH constructs. Light chain for the KIH format consists of the light-chain variable region of cetuximab connected to the human kappa constant region (Figure 1B). Both formats of anti-EGFR/PD-L1 BsAbs were generated and produced by standard DNA recombinant technologies, and BsAbs were purified from cell culture supernatant by protein-A chromatography and analyzed by non-reducing and reducing SDS-PAGE. The predicted molecular weights of the DVD-Ig and KIH formats are 207 kDa and 129 kDa, respectively. Data from SDS-PAGE analysis under the non-reducing condition shows that DVD-Ig yielded a single band at ~220 kDa and the KIH format yielded a major band at 130 kDa and two minor bands at ~100 kDa and 50 kDa. The minor bands at ~100 kDa and 50 kDa appears to be side products of the KIH BsAb format which correspond to the pairing of two half-heavy chains and a single half-heavy chain, respectively, without light chains (Figure 1C). Data from SDS PAGE analysis under reducing conditions shows that the DVD-Ig format of anti-EGFR/PD-L1 BsAbs yielded ~75 kDa heavy-chain and ~37 kDa light-chain bands, whereas the KIH format yielded ~50 kDa heavy-chain and ~25 kDa light-chain bands (Figure 1C).

### 3.2. Binding Characterization of Anti-EGFR/PD-L1 BsAbs

The binding kinetics data obtained from surface plasmon resonance (SPR) measurements using the Biacore optical biosensor instrument indicated that both formats of anti-EGFR/PD-L1 BsAbs retained binding activity towards EGFR and PD-L1. However, the equilibrium dissociation constants (KD) of the KIH format for EGFR and PD-L1 by Biacore were found to be weaker than the DVD-Ig format and the corresponding mAbs, cetuximab and atezolizumab, respectively (Figure 2A). The binding kinetics parameters for the DVD-Ig format for EGFR and PD-L1 were comparable to that of their mAbs counterparts (cetuximab and atezolizumab (Figure 2A)). We observed a higher kd (1/s) rate for the DVD-Ig format, which could be responsible for higher KD values of DVD-Ig compared to cetuximab for EGFR binding. An ELISA binding assay was performed to determine the dose-dependent binding profile of anti-EGFR/PD-L1 BsAbs against immobilized human EGFR and PD-L1, and were compared with Cetux and ATE, respectively. The optical density (OD) data from the ELISA assay indicate that both formats of anti-EGFR/PD-L1 BsAbs bound to EGFR and PD-L1 antigens. The tetravalent DVD-Ig format of BsAbs appears to have stronger binding towards EGFR in comparison to the bivalent KIH format. (Figure 2B). The binding capacity of anti-EGFR/PD-L1 BsAbs to the EGFR and PD-L1 antigens expressed on the cell surface of TNBC cells was assessed by flow cytometric analysis. Median fluorescence intensity (MFI) data obtained from flow cytometric staining show that although both DVD-Ig and KIH formats bound to EGFR and PD-L1 with strong affinity, the comparative analysis indicates that the KIH format has slightly stronger affinity than the DVD-Ig format in all three TNBC cells. (Figure 2C). A coimmunoprecipitation assay using MDA-MB-231 and BT-20 cells was performed, which demonstrated that EGFR and PD-L1 were coimmunoprecipitated with both anti-EGFR/PD-L1 BsAb formats. (Figure 2D). Taken together, these data indicate that different BsAb formats might have slightly different sensitivity towards antigens when assessed with different binding methods; therefore, the selection of appropriate binding analytical methods could be an important consideration to determine the binding activity of different molecular formats of BsAbs.

### 3.3. The Antitumor Potency of BsAbs

To investigate the potencies of anti-EGFR/PD-L1 BsAbs in triple-negative breast cancer (TNBC) cellular models, cell proliferation and cell viability assays were performed. Data from a trypan blue cell proliferation assay showed that the DVD-Ig format, but not the KIH format, significantly inhibited cellular proliferation of MDA-MB-231 cells on day 5 of the treatment compared to the untreated or parental mAb controls (Figure 3A). Both formats of anti-EGFR/PD-L1 BsAbs did not exhibit significant growth inhibition in BT-20 cells, in comparison to the parental and untreated controls in the trypan blue cell proliferation assay (Figure 3A). Cellular proliferation inhibition by anti-EGFR/PD-L1 BsAbs was determined by an orthogonal method using a cell viability assay from Promega in MDA-MB-231 and BT-20 cells. Data from the cell viability assay showed that both formats of anti-EGFR/PD-L1 BsAbs were capable of significantly inhibiting cell viability in MDA-MB-231 cells similar to Cetux treatment (Figure 3B). In BT-20 cells, we observed that the DVD-Ig format was even more potent in inhibiting cellular viability compared to Cetux treatment on day 3 (Figure 3B). Differences in cell growth inhibition data obtained from these two assays could be attributed to differences in assay principles, seeding density, plating format and readout of the two assays. The trypan blue exclusion assay is based on the principle that live cells possess intact cell membranes which exclude trypan blue dye, whereas the Promega CellTiter-Glo assay determines the viability of cells based on ATP content of the metabolically active cells. Overall, these results demonstrate that DVD-Ig and KIH formats exhibit some differences in the inhibitory potency when different cells or assays were used. Interestingly, the cell viability assay could detect the inhibitory potency of DVD-Ig and KIH BsAbs in both MDA-MB-231 and BT-20 cells, whereas the trypan blue cell proliferation assay was not sensitive enough to detect inhibitory properties of BsAbs in BT-20 cells. These data also suggest that the types of cells and assays chosen are essential for assay development and potency testing.

### 3.4. PD-1/PD-L1-Blocking Activity

The PD-1/PD-L1 blockade bioassay assesses the mechanism of action (MOA) of antibody therapeutics that are designed to block PD-1/PD-L1 interaction [26]. The PD-1/PD-L1-blocking activity of anti-EGFR/PD-L1 BsAbs was assessed using a standard PD-1/PD-L1 blockade assay in which CHO-K1 cells were engineered to express human PD-L1 and cell surface protein designed to activate cognate TCRs in an antigen-independent manner. Data from the PD-1/PD-L1 blockade assay showed that both formats of anti-EGFR/PD-L1 BsAbs retained their ability to block PD-1/PD-L1 interaction and have a comparable blocking profile to ATE (Figure 4). As expected, cetuximab did not show any PD-1/PD-L1-blocking activity. These data indicate that both formats of anti-EGFR/PD-L1 BsAbs can disrupt the engagement of PD-1 to its ligand PD-L1 and thereby promote TCR signaling, transcriptional activation and cytokine production.

### 3.5. T Cell-Mediated Killing Assays

T cell-mediated antitumor immune activity of both BsAb formats compared side-by-side using TNBC cells, ATE and Cetux were also included as controls. After labelling the TNBC cells (MDA-MB-231 and BT-20 cells) with calcein AM dye, TNBC cells were co-cultured with activated T cells in the presence or absence of mAbs and BsAbs. Celigo Imaging System was used to count live tumor cells labeled with calcein AM dye. As shown in Figure 5, both DVD-Ig and KIH formats of anti-EGFR/PD-L1 BsAbs significantly enhanced T cell-mediated cytotoxicity of MDA-MB-231 and BT-20 cancer cells when compared with T cells-only incubation. Although ATE treatment also significantly enhanced the killing of cancer cells, no statistically significant difference was observed between ATE and the two formats of anti-EGFR/PD-L1 BsAbs regarding T cell-mediated killing of cancer cells. As expected, Cetux failed to promote the significant killing of cancer cells in the presence of activated T cells. These data indicate that both anti-EGFR/PD-L1 BsAb formats have comparable profiles regarding the T cell-mediated killing of MDA-MB-231 and BT-20 cells, and no statistically significant difference between two formats of BsAb was observed.

### 3.6. Cytokine Production Assays

PD-L1/PD-L1 interaction downregulates T cell function and suppresses the production of pro-inflammatory cytokines such as IFNγ and TNFα. However, disruption of PD-1 and PD-L1 interaction by atezolizumab has been shown to contribute enhanced production of IFNγ and TNFα. In our study, we tested whether anti-EGFR/PD-L1 BsAbs can enhance cytokines IFNγ and TNFα production in the presence or absence of activated T cells. As shown in the Figure 6, co-culture of TNBC cells and T cells in presence of ATE, DVD-Ig and KIH significantly increased the production of IFNγ and TNFα. Comparative analysis of cytokine production data indicate that the DVD-Ig format induced higher production of both cytokines in comparison to KIH. Additionally, the DVD-Ig format of anti-EGFR/PD-L1 BsAbs induced higher production of cytokines when compared with ATE alone. Cetuximab treatment did not significantly enhance the cytokines in both TNBC cells except for TNF production in BT-20 cells. These data also suggest that although the KIH format has slightly weaker binding affinity for PD-L1 and lower cytokine production compared to the DVD-Ig format, it appears that it does not impact PD-L1/PD-1-blocking and T cell-mediated killing activities.

### 3.7. ADCC Activity

The Fc fragment of atezolizumab contains aglycosylated Fc mutation in the heavy chain, which means it cannot bind to Fc receptors and therefore does not have ADCC activity [27]. Conversely, cetuximab can cause ADCC activity in MDA-MB-468 cells [28]. Our anti-EGFR/PD-L1 BsAbs contain the Fc fragment from human IgG1; therefore, we investigated whether the two formats of anti-EGFR/PD-L1 BsAbs can induce ADCC activity in high-EGFR-expressing MDA-MB-468 cells. The data from the ADCC reporter bioassay showed that dose-dependent ADCC activity induced by the DVD-Ig format was similar to that induced by Cetux in MDA-MB-468 cells (Figure 7A). The KIH-format anti-EGFR/PD-L1 BsAb also exhibited ADCC activity, but not as potent as DVD-Ig or Cetux (Figure 7A). As expected, ATE did not show any ADCC activity in our experiment (Figure 7A). We also tested the ADCC activity in MDA-MB-231 cells that express high levels of EGFR, but interestingly, we failed to observe any ADCC activity in MDA-MB-231 cells from BsAbs and Cetux (Figure 7B). Taken together, these data reveal two critical points with regard to the characterization of BsAbs targeting EGFR. First, in addition to other aspects that may affect ADCC activity of BsAbs, the formats of BsAbs may affect the potency of ADCC activity mediated by BsAbs. This may be attributed to the binding affinity of BsAbs for EGFR. Secondly, the results from Figure 7B demonstrated that the cell line chosen for the ADCC assay is critical for the successful detection of ADCC activity mediated by anti-EGFR BsAb, as well as mAbs.

### 3.8. Inhibition of Ligand-Induced EGFR Signaling by Anti-EGFR/PD-L1 BsAbs

The molecular mechanisms of antitumor activities of the anti-EGFR/PD-L1 BsAbs were explored by examining the effect of BsAbs and control mAbs on blocking phosphorylation of EGFR and its downstream targets in TNBC cells. For this purpose, MDA-MB-231, BT-20 and MDA-MB-468 cells were pre-treated with mAbs and BsAbs and then exposed to EGF. As shown the in Figure 8A, the two formats of anti-EGFR/PD-L1 BsAbs eliminated the EGF-induced phosphorylation of EGFR in MDA-MB-231 cells mimicking the cetuximab-mediated effect. The phosphorylation status of Akt, a downstream signaling target of EGFR, was also blocked by the DVD-Ig and KIH formats of anti-EGFR/PD-L1 BsAbs, which was similar to cetuximab-mediated effects (Figure 8A). Anti-EGFR/PD-L1 BsAbs also downregulated the EGF-induced phosphorylation effects in BT-20 and MDA-MB-468 cells which were comparable to cetuximab treatments (Figure 8B,C). ATE did not show any effect on EGF-induced phosphorylation of EGFR or its downstream targets in any of the TNBC cells tested in this study (Figure 8A–C). These results indicate that the Cetux arm of anti-EGFR/PD-L1 BsAbs maintained its capability to effectively block EGFR-mediated signaling in TNBC cells.

### 3.9. Antitumor Activity of Anti-EGFR/PD-L1 BsAbs in Tumor Xenograft Model

An in vivo tumor xenograft study using athymic nude mice was performed to determine antitumor activity of two formats of anti-EGFR/PD-L1 BsAbs compared to parental mAbs. TNBC cell line MDA-MB-231 was subcutaneously injected into the flank region of mice and when tumor volume reached 50–100 mm^3^, anti-EGFR/PD-L1 BsAbs, cetuximab, atezolizumab or saline was injected in tumor-bearing mice. Figure 9A shows the images of excised tumors at the end of the treatment course. Data from tumor calculations as shown in Figure 9B indicate that both formats of anti-EGFR/PD-L1 BsAb significantly inhibited tumor growth compared to the saline-treated mice group. Figure 9C,D represents the tumor growth curve and mice body weight measurements during treatment. Although anti-EGFR/PD-L1 BsAbs were equally potent in suppressing tumor growth as corresponding mAbs, no significant tumor difference in growth inhibition was observed between anti-EGFR/PD-L1 BsAbs and corresponding mAbs.

## 4. Discussion

KIH technology has been widely utilized to improve Fc heterodimerization and reduce the number of non-functional combinations of heavy-chain assembly. The modified KIH structure, which is generated by creating a “knob” in one heavy chain and a “hole” in another heavy chain, promotes the correct assembly between heavy chains from different mAbs and provides for more than 90% of correctly assembled products under co-expression conditions. The stability of KIH heterodimers was studied in comparison to hole-hole homodimer variants, which demonstrate that KIH heterodimers have very stable conformation, thus supporting the knobs-into-holes heterodimerization as a rational BsAb design strategy [29]. Another study using anti-CD20 antibody revealed that ADCC activity is retained in KIH-generated BsAbs, and half the asymmetric heterodimer is sufficient to produce ADCC enhancement similar to that observed for a fully afucosylated anti-CD20 antibody indicating that KIH-based BsAbs retain Fc effector functions [30]. On the other hand, symmetrical tetravalent BsAb molecules can be generated by the DVD-Ig approach. The DVD-Ig molecule preserves the dual antigen-binding specificity from both corresponding mAbs and offers structural stability for large-scale production [31,32]. In this investigation, we employed both approaches to generate anti-EGFR/PD-L1 BsAbs using the antigen-binding domains of cetuximab and atezolizumab and performed a side-by-side characterization of both formats of BsAbs. Our binding data from the Biacore experiment suggest that although both formats bind to their respective antigens, the KIH format has slightly weaker binding affinity for both EGFR and PD-L1, compared to the DVD-Ig format. Comparative analysis of results from most of our experiments indicates that although both formats have the capability to induce an antitumor response in TNBC cells, the DVD-Ig format of anti-EGFR/PD-L1 BsAbs appears to offer slight advantage over the KIH format. The tetravalent structure of DVD-Ig offers two binding sites for each antigen, whereas the bivalent nature of KIH format offers one binding site for each antigen.

Cetuximab exhibits its antitumor immune response through activating multiple mechanisms [33]. ADCC activity of cetuximab is mediated by NK cells in which the Fc portion of cetuximab IgG binds to CD16/FcγRIIIa receptors on NK cells resulting in NK cell activation [12]. Activated NK cells induce lytic activity on tumor cells and release tumor antigens. These tumor antigens can be cross presented by dendritic cells to cytotoxic T cells which can prime them for additional tumor cell killing [34]. Furthermore, cetuximab-mediated NK cells can also release IFN and other cytokines in the intertumoral space which can facilitate crosstalk between dendritic cells, macrophages and other immune cells. Cytokine-mediated crosstalk among the immune cells can lead to infiltration of cytotoxic T cells in intra-tumoral space, thus providing a long-term immune response [34]. However, cetuximab-mediated immune activity can be dampened by stimulation of counter-regulatory immunosuppressive feedback loops involving T regulatory (Treg) cells and myeloid-derived suppressor cells (MDSC) [33,34]. The Treg and MDSC-mediated immunosuppressive mechanism depletes expression of ADCC markers (e.g., perforin and granzyme) and upregulates immune checkpoints on tumors and immune cells. Therefore, synergizing the cetuximab-mediated ADCC activity with PD-L1/PD-1-blocking immune activity can fully mobilize the adaptive and innate immune systems against tumor cells [33]. Our data show that the DVD-Ig format function was similar to cetuximab in stimulating ADCC activity against MDA-MB-468 cells and demonstrated a comparable profile for blocking PD-L1/PD-1 axes, thus providing dual immune functionality using a single bispecific molecule. We also observed that our BsAbs failed to show ADCC activity in MDA-MB-231 cells. It should be noted that the ADCC Reporter Bioassay does not measure actual lytic activity of effector cells; instead, it uses an alternative readout at an earlier point in ADCC pathway activation: the activation of gene transcription through the NFAT (nuclear factor of activated T cells) pathway in the effector cell. Actual lytic ADCC assays may give different results in MDA-MB-231 cells than ADCC reporter assays. Furthermore, stably expressing engineered Jurkat cells utilized in this ADCC Reporter Bioassay may also affect the expression of EGFR on the MDA-MB-231 cell surface, which in turn might impact the ADCC readout. Moreover, it has been shown in the literature that tumor cells may develop ADCC resistance due to genetic and epigenetic changes that lead to a general loss of target cell adhesion properties that are required for the establishment of an immune synapse, killer cell activation and target cell cytotoxicity [35]. Finally, MDA-MB-231 cells express both EGFR and PD-L1 at a higher level. Co-expression of two antigens on the same target cell (MDA-MB-231) could also impact the ADCC activity induced by BsAbs that targets those two antigens. Our data indicate that for BsAbs with a functional Fc, multiple cell lines may need to be included for the characterization of ADCC activity to ensure adequate understanding of the mechanism of action that contributes to antitumor activity of BsAbs in vivo. As a result, the potency of ADCC activity of BsAb should be monitored and controlled during clinical study.

PD-L1-blocking agents including atezolizumab have been extensively investigated for treatment of tumor microenvironments of advanced and metastatic TNBC. TNBC tumors are remarkably heterogenous and are comprised of not only tumors cells, but also multiple other cell types including tumor-infiltrating immune cells, T cells and macrophages [36]. The immunogenic nature of TNBC, as evidenced by a high number of tumor-infiltrating cells and high PD-L1, provides a strong rationale for testing immune checkpoint-blocking therapies; however, currently available data from clinical studies suggest that single-agent immunotherapies have only moderate response rate. Furthermore, the discrepancy in the results of two atezolizumab clinical trials started discussions in the oncology community about the efficacy of atezolizumab monotherapy in treating metastatic TNBC [37]. Hence, co-targeting different components of TNBC tumor microenvironments could provide a more comprehensive approach to enhance overall antitumor activity. Atezolizumab has high binding affinity toward human and mouse PD-L1 [38] and could potentially disrupt mouse PD-1/PD-L1 interaction [39]. Thus, the mouse is a relevant animal model to study anti-PD-L1 mAbs and BsAbs targeting EGFR/PD-L1. Immunodeficient nude mice models have been utilized to study the antitumor properties of atezolizumab in osteosarcoma and ovarian cancer xenograft models [40,41]. Antitumor effects elicited by atezolizumab in immunodeficient mice models can be attributed to suppression of tumor growth without involvement of the immune system. Recent studies have shown that anti-PD-L1 antibodies can inhibit tumor-intrinsic PD-L1-mediated signaling and suppress tumor growth progression in immunodeficient mice models independent of T cell activation [42,43]. One possible hypothesis could be that the design of our tumor xenograft model may have some limitations to detect the synergistic antitumor activity of BsAbs as compared to mAbs. In this investigation, we performed a side-by-side comparative characterization of the BsAbs with two molecular formats that are commonly used for clinical application. Our results indicate that our BsAbs have the capability to release PD-1/PD-L1 checkpoint inhibition, to induce EGFR-directed tumor anti-proliferative activity and to maintain Fc-mediated ADCC effector functions to provide more pronounced killing of TNBC tumors, suggesting that the BsAbs targeting EGFR and PD-L1 warrant further investigation as a targeted antibody therapeutic to treat TNBC and tumors expressing EGFR.

## Figures and Tables

**Figure 1 pharmaceutics-14-01381-f001:**
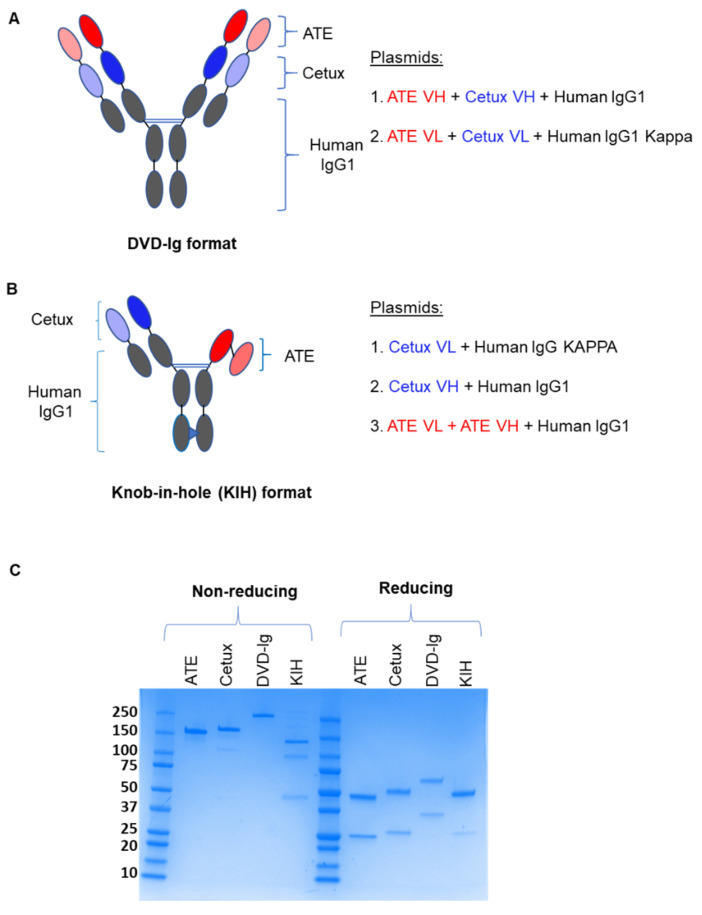
Construction and schematic representation of anti-EGFR/PD-L1 BsAbs: (**A**) Dual variable domain immunoglobulin (DVD-Ig) symmetric tetravalent format of anti-EGFR/PD-L1 BsAb in which single-chain variable fragments of ATE and Cetux are connected to the human IgG Fc portion. (**B**) Knob-in-hole (KIH) asymmetric bivalent format of anti-EGFR/PD-L1 BsAb in which two half-heavy chains containing “knob” mutations (T350V_L351Y_F405A_Y407V) and “hole” mutations (T366L_K392l_T394W) were generated. scFv ATE (light and dark red); scFv Cetux (light and dark blue); human Fc-IgG1 (grey). (**C**) SDS-PAGE analysis showing the heavy and light chains of DVD-Ig and KIH formats of purified anti-EGFR/PD-L1 BsAbs, ATE and Cetux under non-reducing and reducing conditions.

**Figure 2 pharmaceutics-14-01381-f002:**
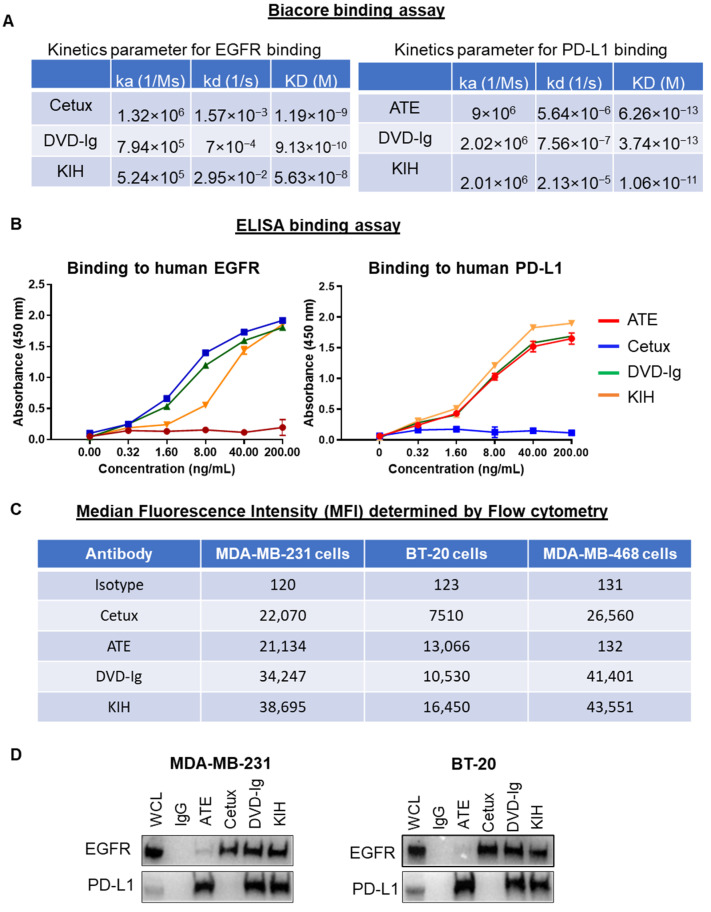
Binding activities of anti-EGFR/PD-L1 BsAbs: (**A**) Binding kinetics parameters of mAbs and BsAbs to PD-L1 and EGFR antigens as detected by a Biacore T200 optical biosensor instrument. (**B**) Dose-dependent binding activity of mAbs and BsAbs using ELISA binding assay showing that anti-EGFR/PD-L1 BsAbs bind to both EGFR and PD-L1 comparable with parental mAbs. (**C**) Binding profile of anti-EGFR/PD-L1 BsAbs and parental mAbs with cell surface EGFR and PD-L1 expressed on MDA-MB-231, BT-20, MDA-MB-468 cells was determined by flow cytometry analysis. Antibody concentration used in this experiment was 1 µg per 1 million cells in 100 µL volume of FACS buffer (1% FBS in PBS). Median florescence intensity (MFI) obtained from flow cytometric staining analysis was presented in tabular format. Human IgG was used as isotype control. (**D**) Coimmunoprecipitation assay using whole-cell lysate of MDA-MB-231 and BT-20 cells to assess the binding of anti-EGFR/PD-L1 BsAbs and parentals mAbs with EGFR and PD-L1. Unprocessed western blots images of Figure 2D are provided in Appendix A.

**Figure 3 pharmaceutics-14-01381-f003:**
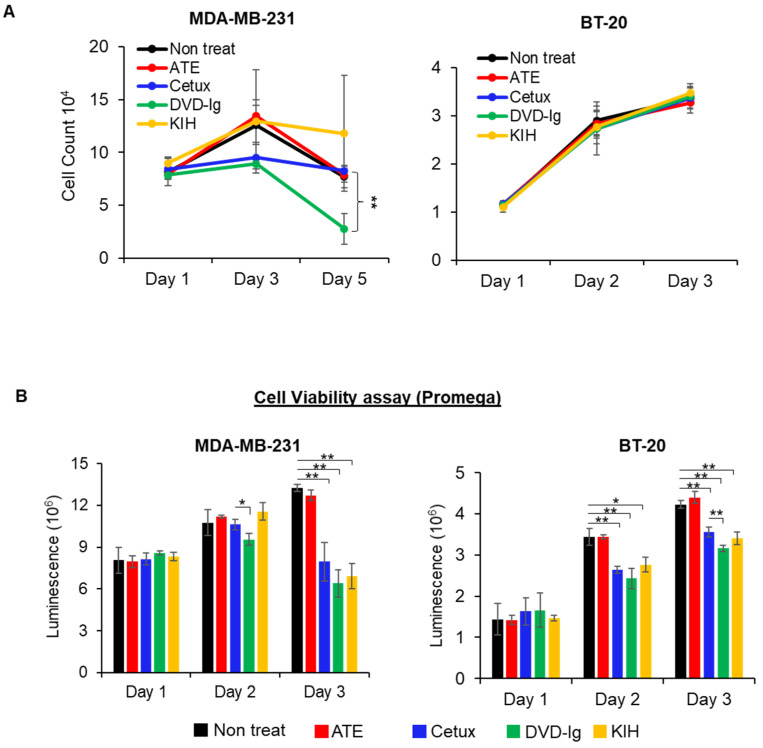
Antitumor potency of BsAbs in TNBC cells. (**A**) Cell proliferation assay was performed in MDA-MB-231 and BT-20 cells after treatment with anti-EGFR/PD-L1 BsAbs and parental mAbs. After treatments, live cells in non-treated and treatment groups were counted using trypan blue exclusion method at indicated timepoints. (**B**) CellTiter-Glo luminescent cell viability assay was performed in MDA-MB-231 and BT-20 cells. For this experiment, 10,000 cells were seeded in white-bottom 96-well plates and allowed to adhere overnight. After treatments with mAbs and BsAbs (20 µg/mL for Cetux, ATE and DVD-Ig, and 40 µg/mL for KIH format of BsAb) for indicated time, CellTiter-Glo reagent was added into the plates to measure luminescence using the Promega Glomax Discover plate reader. Both assays were performed in biological triplicates, and data are representative of two or more experiments. The differences between two groups were considered statistically significant when *p* < 0.05 (*, *p* < 0.05; **, *p* < 0.01).

**Figure 4 pharmaceutics-14-01381-f004:**
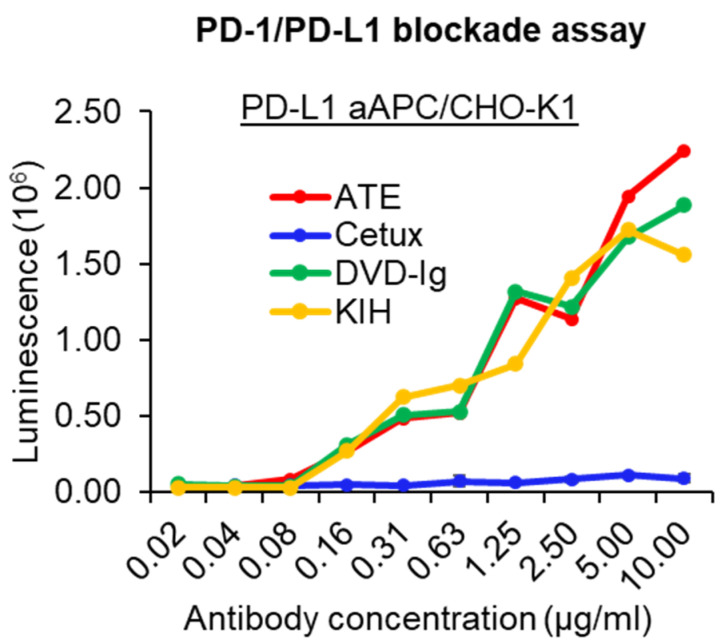
PD-1/PD-L1-blocking activity of anti-EGFR/PD-L1 BsAbs was determined by PD-1/PD-L1 blockade cell-based assay (Promega) to assess the ability of the anti-EGFR/PD-L1 BsAbs to block PD-1/PD-L1 engagement compared to parental mAbs. This experiment has been performed in biological triplicates, and data are representative of two or more independent experiments.

**Figure 5 pharmaceutics-14-01381-f005:**
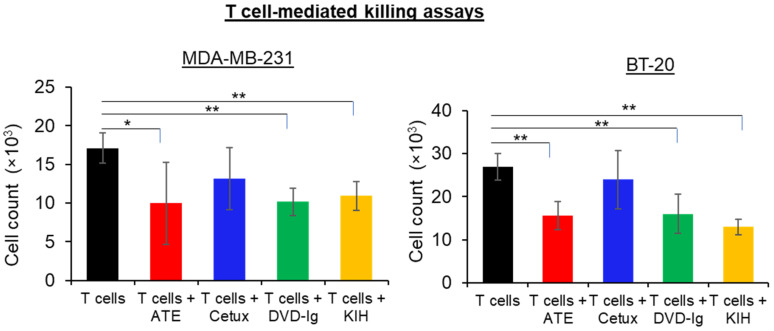
T cell-mediated killing assay was performed in MDA-MB-231 and BT-20 cells, in which calcein AM-labeled cancer cells were co-cultured with activated T cells, then treated with anti-EGFR/PD-L1 BsAbs, parentals mAbs or left untreated for 24 h. After 24 h of incubation, calcein AM-labeled live cells were counted by Celigo Imaging system. T cell-mediated killing assay was performed in triplicates, and data are representative of two or more independent experiments. The differences between two groups were considered statistically significant when *p* < 0.05 (*, *p* < 0.05; **, *p* < 0.01).

**Figure 6 pharmaceutics-14-01381-f006:**
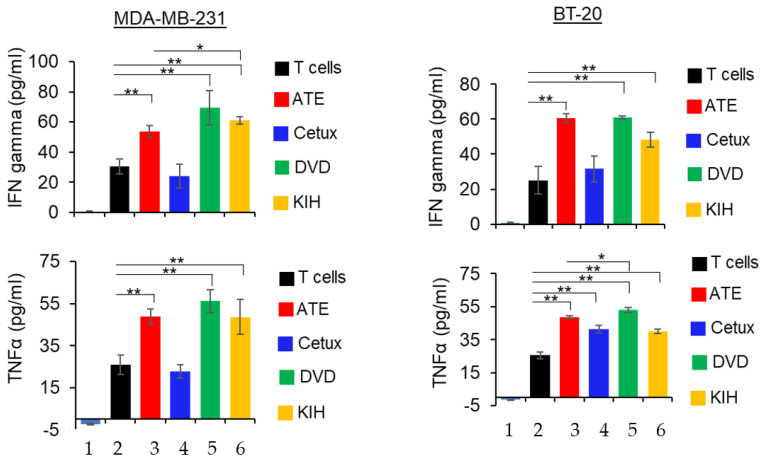
Detection of IFNγ and TNFα in co-culture of cancer cells and T cells in presence of parental mAbs and BsAbs. After 24 h of incubation, the supernatant was collected and then subjected to ELISA assay kit (R and D systems) to detect IFNγ and TNFα. Bar# 1 of each bar graph represents tumor cells only. Bar# 2,3,4,5,6 of each bar graph represents tumor cells co-cultured with T cells. Bar# 3,4,5,6 represents antibody treatments indicated in color codes. This experiment has been performed in biological triplicates, and data are representative of two or more independent experiments. The differences between two groups were considered statistically significant when *p* < 0.05 (*, *p* < 0.05; **, *p* < 0.01).

**Figure 7 pharmaceutics-14-01381-f007:**
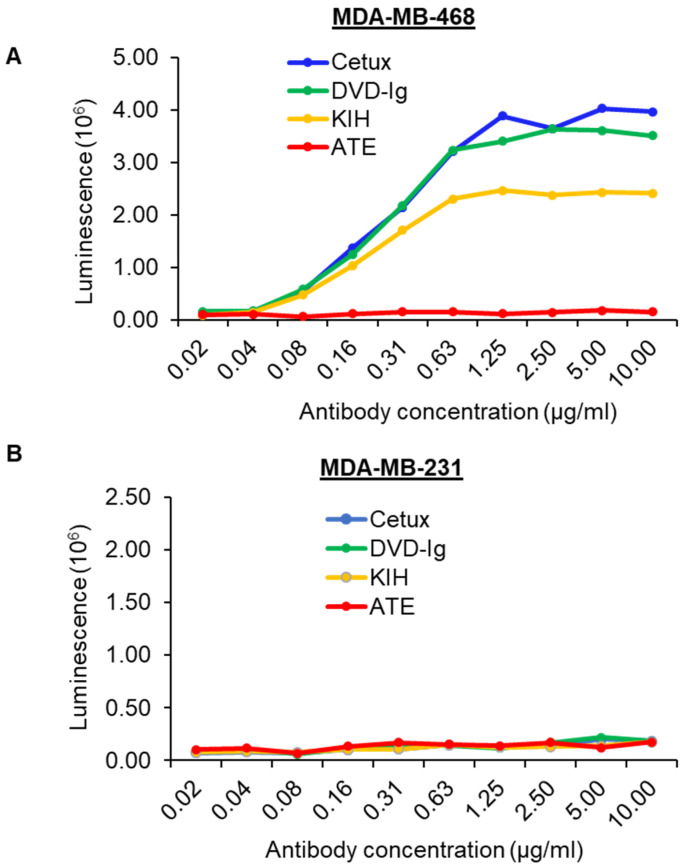
ADCC activity induced by anti-EGFR/PD-L1 BsAbs. ADCC activity was determined by ADCC Reporter Bioassay. Target cancer cells MDA-MB-468 (**A**) and MDA-MB-231 (**B**) were seeded in 96-well plates and incubated with serially diluted antibodies and effector cells for 5–6 h. Bio-Glo™ Luciferase Assay Reagent was added to the plates and luminescence was measured using GloMax discover plate reader.

**Figure 8 pharmaceutics-14-01381-f008:**
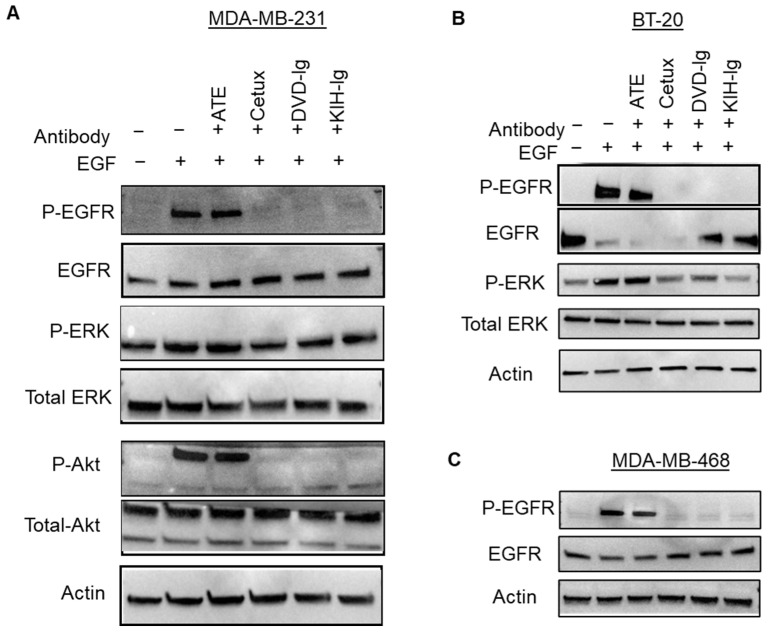
Anti-EGFR/PD-L1 BsAbs inhibited ligand-induced signaling in TNBC cells. MDA-MB-231 (**A**), BT-20 (**B**) and MDA-MB-468 (**C**) cells were subjected to serum starvation overnight and then pre-treated with anti-EGFR/PD-L1 BsAbs and parental mAbs. After antibody treatment, cells were exposed to EGF for 30 min at a concentration of 50 ng/mL. After treatments, cells were lysed in lysis buffer to prepare whole-cell lysate (WCL), and samples were subjected to western blotting to determine the phosphorylation levels and total protein levels of indicated signaling molecules. Actin expression was performed to measure equal loading. Unprocessed western blots images of (**A**–**C**) are provided in Appendix A.

**Figure 9 pharmaceutics-14-01381-f009:**
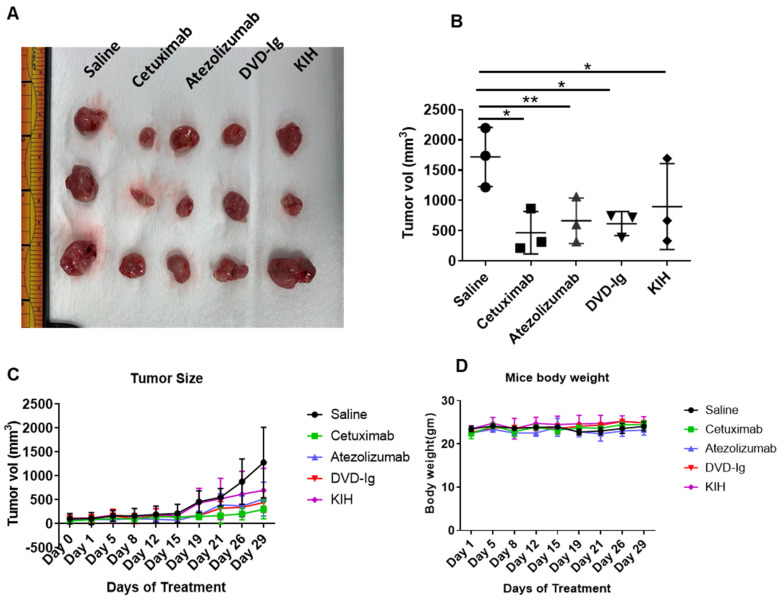
Anti-EGFR/PD-L1 BsAbs exhibited potent antitumor activity in MDA-MB-231 tumor xenograft study. Five million MDA-MB-231 cells were injected in left flank of athymic nude mice. Treatment conditions: saline, ATE, Cetux and anti-EGFR/PD-L1 BsAbs were given intraperitoneally at 10 mg/kg body weight twice a week. (**A**) The images of the excised tumors at the end of the treatments. (**B**) Tumor volume was calculated after removing the tumors from mice at the end of treatment course on day 29; *, *p* < 0.05; **, *p* < 0.01. (**C**) Tumor growth curve over the course of treatment. (**D**) Mouse body weights in each treatment group were measured over the course of treatment.

## Data Availability

The data presented in this study are available in the article and Appendix A.

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
