# Peer review of "Comparative Characterization of Different Molecular Formats of Bispecific Antibodies Targeting EGFR and PD-L1"

_pharmaceutics, 2022, doi:10.3390/pharmaceutics14071381_

Round 1

Reviewer 1 Report

The manuscript by Dr. Mohan et al. describes their work on characterization of two different bispecific antibodies targeting EGFR and PD-L1 in either symmetrical DVD-Ig and asymmetrical knob-in-hole (KIH) formats. They showed both formats bound EGFR and PD-L1 from Biacore, ELISA, and FACS analysis. The results from in vitro analysis demonstrate anti-tumor proliferation although the potencies vary dependent on different cells and assays used. The in vivo anti-tumor activity using nude mice was also presented. The work is interesting and straightforward. However, the manuscript needs to be revised before being considered for publication in the journal. I list a few points for consideration as bellow.

1. I don’t see any description of in vivo work in the abstract although the in vivo data has been provided (Figure 9).

2. I am wondering whether the authors have done any other biophysical studies, such as aggregation and stability of these two different bispecific antibodies. Those parameters may have impact on their data interpretation.

3. Based on the data, it seems to me that the results from most of anti-tumor potencies correlated well with the affinity measured by SPR (higher affinity binding of DVD-Ig than KIH). The difference in binding of two bispecific antibody formats from ELISA could be due to variation from the assay itself.

4. The increased affinity (KD) of DVD-Ig to either EGFR or PD-L1 (as compared to monoclonal antibodies cetuximab or atezolizumab) seems to be mainly due to an increase in kd (1/s) (or koff rate). It would be better to mention that.

5. Is it possible to show simultaneous binding of these two bispecific antibodies to EGFR and PD-L1 on single cell? The interaction may be testable using imaging methods.

6. It is interesting that there was great difference in ADCC activities when two different tumor cell lines were used. Is there any difference in cell culture components for these two cell lines? Since MDA-MB-468 385 cells have much less PD-L1 on cell surface, does that mean some impact from PD-L1 on the activity? With bispecific antibodies, does the simultaneous binding of bispecific antibodies to both antigens on the cells interfere with interaction of the Fc to FcγRIIIA?

7. The authors should discuss why there was significant difference in the ADCC activities of their bispecific antibodies when two different target cell lines were used.

8. I don’t think that tumor xenograft study using athymic nude mice is very relevant to the current studies for blocking PD-L1 mediated checkpoint inhibition. The anti-tumor activity may be only due to anti-proliferation through EGFR inhibition. The authors should discuss that in the discussion section.

9. Other minor points:

a. There is a mistake from original references of atezolizumab related to N298A mutation as cited in this manuscript (line 381, page 11). It should be a mutation at N297A since native Fc contain N at 297 while S at 298 based on Eu numbering. It may be better to mention as aglycosylated Fc mutation instead of “asparagine-to-alanine substitution at position 298 in each CH2 domains”?

b. I don’t understand the sentence “It should be noted that atezolizumab has high binding affinity toward human and mouse PD-L1 [29] and mouse PD-1 binds to mouse PD-L1 [30]” (lines 436 to 438, pages 13 and 14). What does it mean for mentioning “mouse PD-1 binds to mouse PD-L1”?

c. “the KIH format has slightly weaker binding affinity for both, EGFR and PD-L1, compared to DVD-Ig format” (lines 483 to 485, page 15) should be “the KIH format has slightly weaker binding affinity for both EGFR and PD-L1, compared to DVD-Ig format” (no comma before EGFR).

Reviewer 2 Report

The study of  Mohan et al, compares two bispecific antibody formats targeting PDL-1 and EGFR for potential synergism of anti-tumor activity in triple negative breast cancer. Overall, the study is well conducted, showing only few differences of the activity of the 2 formats. Some results lack significativity for rigorous comparison. This study will be useful for the antibody engineering community but I suggest that more details should be given for their characterization (see suggestions below). If the 2 formats are equivalent in their activity, the choice of one of them for further development should rely on developpability, which is not enough addressed in this study.

Lane 104, give details on the vectors used for expression, give détails on the yeild per mL of supernatant obtained for each construct

Lane 171 : give the origin of the T cells used for the T-cell mediated killing assay (or reference)

Lane 179 : give details (or reference)  on the method used for activation of T cell (or lane 342)

Result section :

Lane 224 : single chain heavy variable fragments,

Lane 233-235 : justify the choice of the mutations used for the knob and the hole variant (or reference)

Figure 1C : add a SDS-PAGE in non reducing conditions to detect potential aggregates and format heterogeneity.

lane 263-265 : explanation of figure 2B : avoid the term « comparable »,  and comment the data regarding monovalency/bivalency of the formats.

Lane 271 : /figure 2C : indicate the concentration of Ab used for the binding test by cytometry

Lane 273 : « demonstrated that anti-EGFR/PD-L1 BsAbs were co-immunoprecipitated with both EGFR 273 and PD-L1 », change into ; « demonstrated that EGFR and PD-L1 were co-immunoprecipitated with both anti-EGFR/PD-L1 BsAbs »,

Figure 2B : it is intriguing that the KD for EGFR  of the KIH format (monovalent) compared to DVD and CX (bivalent) is not higher. It causes the interrogation that the KIH format might be composed also of a significant part of EGFR bivalent species.  

Figure 2C/2D : in WB figure 2D, BT-20 seems to express more PD-L1 than MDA-MB-231 cell line for a similar level of PDL1 expression.  No coherent with surface expression by FACS. Please, explain the figure annotation for 2D lane 288.

Lane 312 ; suggest an hypothesis for explaining the different output obtained with the viability tests on the 2 cell lines ? Figure 3 : use the same representation for the 2 tests panel A and B to facilitate comparison. Indicate statistics the same way. Indicate if experiment has been done more than once.

Lane 351 : indicate the concentration of Ab used for rigourous comparison when comparing formats.

Figure 4 : indicate if the experiment has been done more than once. I would suggest to plot Ab molar concentration (instead of mass concentration) for better comparison

Figure 5 : precise the concentration of Ab used in the T cell killing assay. Compare T cell mediated  killing effect of the BsAb with the conditions with CX  only of PD-L1 only.

Lane 364/figure 6 : I cannot see the condition « in absence of T cells ». Use the same scale for Y axis for both cell lines. Indicate similar statistic comparison on all the graph. Indicate how many time was done the experiment (biological replicates).

Lane 367-368-370 : no statistical difference is seen when comparing ATE and BsAb effect on the cytokine production by T cells as quoted. No statistical différence is seen when comparing both BsAb as quote.

Lane 389, lane 396 : the lower activity fo KIH maybe due to monovalent binding . at some poitn if would be

Lane 391 : what is the hypothesis of the resistance mechanism to ADCC of MDA MB 231 cells  figure 7B, give an hypothesis.

Paragraph 3.9/figure 9 : this paragraph concerns the in vivo study. This paragraph should not contain elements of discussion (lane 448-454 should be removed, reference 33, 34 should be commented in the discussion part but not in the result section and the hypothesis of the mechanism of ATE effect in this model shoud be clearly exposed ). Figure 9C should show some statistics .

From lane 464 : the discussion is quite weak, it  should focus on the comparison of the effects of the formats based on the results obtained in the study.  If the 2 formats are equivalent, the choice of one of the for further development should rely on developpability, which is not addressed in this study.
